# Entropy of Real-World Gait in Parkinson’s Disease Determined from Wearable Sensors as a Digital Marker of Altered Ambulatory Behavior

**DOI:** 10.3390/s20092631

**Published:** 2020-05-05

**Authors:** Lucy Coates, Jian Shi, Lynn Rochester, Silvia Del Din, Annette Pantall

**Affiliations:** 1School of Mathematics, Statistics and Physics, Newcastle University, Newcastle upon Tyne NE1 7RU, UK; Lucy.A.Coates@hotmail.com (L.C.); Jian.Shi@newcastle.ac.uk (J.S.); 2Translational and Clinical Research Unit, Clinical Ageing Research Unit, Newcastle University, Newcastle upon Tyne NE4 5PL, UK; Lynn.Rochester@newcastle.ac.uk (L.R.); Silvia.Del-Din@newcastle.ac.uk (S.D.D.); 3Newcastle upon Tyne Hospitals NHS Foundation Trust, Newcastle upon Tyne NE7 7DN, UK

**Keywords:** wearable technology, gait, Parkinson’s disease, sample entropy, variability, real-world

## Abstract

Parkinson’s disease (PD) is a common age-related neurodegenerative disease. Gait impairment is frequent in the later stages of PD contributing to reduced mobility and quality of life. Digital biomarkers such as gait velocity and step length are predictors of motor and cognitive decline in PD. Additional gait parameters may describe different aspects of gait and motor control in PD. Sample entropy (SampEnt), a measure of signal predictability, is a nonlinear approach that quantifies regularity of a signal. This study investigated SampEnt as a potential biomarker for PD and disease duration. Real-world gait data over a seven-day period were collected using an accelerometer (Axivity AX3, York, UK) placed on the low back and gait metrics extracted. SampEnt was determined for the stride time, with vector length and threshold parameters optimized. People with PD had higher stride time SampEnt compared to older adults, indicating reduced gait regularity. The range of SampEnt increased over 36 months for the PD group, although the mean value did not change. SampEnt was associated with dopaminergic medication dose but not with clinical motor scores. In conclusion, this pilot study indicates that SampEnt from real-world data may be a useful parameter reflecting clinical status although further research is needed involving larger populations.

## 1. Introduction

Advances in medicine and public health preventive strategies contribute to the increased life expectancy of an expanding world population. However, a consequence of greater longevity is increased morbidity, resulting in the loss of independent living, frailty, and mortality [1]. There are substantial health and social costs linked to the loss of independence as well as the impact on quality of life of the individual and careers. A common age-related neurodegenerative disease is Parkinson’s disease (PD), with a UK incidence of 84 per 100,000 in adults over 50 years [2]. In PD, deep brain structures controlling movement degenerate and there is progressive loss of motor function. Symptoms of PD include gait impairment, associated with an increased risk of falling, which is a major health concern and economic burden [3]. The quantitative analysis of gait may provide biomarkers of disease progression and offer insight into motor control strategies. Wearable technology, such as accelerometers, allow for monitoring of ambulatory activity in both controlled and real-world environments [4]. Therefore, data can be collected over an extended period of time when the participant is walking in familiar surroundings, which provides a more accurate representation of gait compared to walking in a gait laboratory [5]. Additionally, analyzing gait through accelerometers offers an inexpensive and portable solution compared to costly clinical laboratory based measurements [6]. Real-world gait recordings, however, present several challenges. Firstly, the environment is unconstrained, unlike a clinical gait laboratory. The terrain will vary, unanticipated obstacles may appear such as vehicles when crossing the road and frequently the person will perform additional tasks, all of which will modify gait patterns. However, if the purpose of real-world gait analysis is to capture the individuals’ gait patterns, this cannot be adequately assessed within the context of an artificial framework but must incorporate the richness of the environment with which the individual engages. 

An essential element when processing large data sets is selecting appropriate features that encapsulate gait information over extended time-periods. A real-world recording from the triaxial AX3 accelerometer (Axivity, York, UK) set at 10 bits resolution, sampling at 100 Hz, generates up to 250 MB of raw binary data over seven days. Digital outcomes from real-world recordings include macro gait measures of walking activity or micro gait measures of specific gait parameters such as gait velocity, stride time, or step length [7]. Stride time is a measure of rhythmicity and by inference automaticity of movement [8], which has been extensively examined in people with PD. Rhythmicity or regularity of stride time can be quantified by linear measures of variability such as the coefficient of variation (CV). Studies have reported that stride time CV is greater for people with PD compared to healthy older adults [9], and for freezers compared to nonfreezers [10]. Stride time CV decreases following dopaminergic medication and is lower in nonfallers compared to fallers [11]. Stride time CV in healthy adults has also been correlated with cortical activity (supplementary motor area, precentral gyrus) [12]. Therefore, the variability of stride time would appear as a good indicator of PD and a risk factor for falls. However, CV and other linear measures of variability are limited as processing involves averaging values over the recording duration, which does not reflect the natural fluctuations in metrics over time. This intrinsic signal variability may relate to underlying motor regulation systems. Nonlinear methods such as entropy, maximum Lyapunov exponent, autocorrelation or recurrence fluctuation analysis provide a measure of temporal variability and may indicate the integrity of the underlying motor control system. Two studies on nonlinear variability of stride time, using detrended fluctuation analysis and long range autocorrelation (LRA), reported more random variability for stride time in people with PD compared to healthy controls [13,14]. By contrast, Kamath (2015) reported that the regularity of stride time, assessed with sample entropy (SampEnt), was greater in people with PD compared to older adults (HOA) [15]. Warlop et al. (2016) reported that a lower Hurst exponent and α-exponent derived from LRA, indicating more random gait, was associated with a greater disease level of PD [16]. 

The application of entropy algorithms to biomedical signals and gait is a common method of nonlinear analysis [17,18,19,20,21]. Entropy measures the probability that a similar pattern in a signal is repeated and will be followed by additional similar patterns thus, indicating regularity of the time series [22]. A low value of entropy is indicative of greater regularity whereas a high value suggests a time series that is to an extent random. Approximate entropy [23] and SampEnt [24] are two types of entropy measurements. However, approximate entropy has greater bias towards regularity, more sensitivity to parameter choices, and less relative consistency compared to SampEnt. SampEnt is a robust to low-level noise, and finite for both stochastic and deterministic processes [22]. Therefore, SampEnt of gait may be a potential biomarker of PD in addition to indicating disease progression of a disease. The aims of this pilot study were to: (i) Evaluate the feasibility of evaluating SampEnt on real-world accelerometry data; (ii) compare real-world SampEnt in PD with respect to HOA; and (iii) investigate longitudinal changes of SampEnt over 36 months. The hypotheses were that regularity would be lower (greater SampEnt) in people with PD compared to HOA and that it would decrease over three years with increasing neurodegeneration. 

## 2. Materials and Methods 

### 2.1. Participants

A subset of participants from the Incidence of Cognitive Impairment in Cohorts with Longitudinal Evaluation-Gait (ICICLE-GAIT) study [25], which commenced in June 2009, was identified. Ten participants were selected: Five with PD and five HOAs. Participants were excluded if they had any neurological (other than PD), orthopedic, or cardiothoracic conditions that may affect their walking or safety during the testing. Participants with PD were diagnosed with an idiopathic PD according to the UK Parkinson’s Disease Brain Bank criteria [26].

### 2.2. Ethics and Consent

The study was approved by the Newcastle and North Tyneside research ethics committee (ICICLE-GAIT 09/H0906/82) and conducted according to the declaration of Helsinki. All participants signed an informed consent form prior to testing.

### 2.3. Demographic and Clinical Measures

At each assessment time-point, participants attended the Clinical Ageing Research Unit, Newcastle University where demographic information was collected, and clinical testing was performed. Demographic measures included height, weight, gender, and age of the participant. A trained clinical examiner rated the severity of PD on the Motor Section (III) of the Movement Disorders Society-Unified Parkinson’s Disease Rating Scale (MDS-UPDRS-III), approximately 1 h after dopaminergic medication intake, when the medication generally has its peak clinical effect. The levodopa equivalent daily dose (LEDD) was also recorded [27], which is a measure of the most commonly prescribed medication for PD.

### 2.4. Real-World Data: Equipment and Procedure

Real-world gait data were collected using a single tri-axial accelerometer-based body worn monitor (BWM) (Axivity AX3, York, UK; dimensions: 23.0 mm × 32.5 mm × 7.6 mm; weight: 11 g; accuracy: 20 parts per million) (Figure 1). This device has previously been validated for recording a high resolution human movement [28]. The BWM was positioned on the skin overlying the fifth lumbar vertebra with a hydrogel adhesive and covered with a Hypafix (BSN Medical Limited, Hull, UK) (Figure 1). The sampling frequency was 100 Hz and the range ± 8 g. The BWM recorded ambulatory activity continuously for seven days. Seven-day accelerometry data were collected at three time-points 18 months apart (time-points TP1, TP2 = TP1 + 18 months, TP3 = TP2 + 18 months). TP1 was approximately 36 months after initial diagnosis of PD for the PD group. Participants were asked to carry out their daily activities as usual and not to alter any routines. On completion of the seven days, participants returned the device by post to the researchers [7]. 

### 2.5. Data Processing and Analysis

The data from the BWM were downloaded and analyzed with a custom-written MATLAB program, which has previously been validated [4,29]. The data contained all activity recorded during the seven days including sitting, climbing stairs, gardening, and walking. In brief, data were segmented by calendar day and axes transformed to orthogonal gravitational vertical, antero-posterior, and medio lateral axes [30]. Individual ambulatory bouts were identified [31,32,33,34,35,36] and further processed. This method has been validated by comparing gait parameters derived from accelerometry data with parameters calculated from a pressure sensitive mat and wearable body-mounted camera [4,29,32]. Figure 2 shows a segment of BWM output from a real-world data recording.

Thresholds for activity bout length and stride time were applied to all participants‘ data-sets. Shorter activity bouts have a greater stride time variability than longer bouts, although their respective means are similar, centered at approximately 1.2 s (Figure 3). Only bouts of activity between 30 and 60 s were included in the analysis. An inclusion range of 0.5–2.5 s (walking bouts of 12–120 strides) was applied to the parameter stride time [37] as strides outside these times are atypical.

### 2.6. Calculation of Sample Entropy

Given a time series of *N* points, SampEnt is defined as the negative natural logarithm of the conditional probability that vectors of length *m* and *m* + 1 are repeated, with tolerance, *r*, and no self-matches [11]. The distance between these repeated vectors, denoted as *d*[*y_m_*(*j*), *y_m_*(*k*)], is the maximum of the absolute distance between corresponding scalar components [38]. Let B be the number of vector pairs that satisfy:(1)$$d\left[y_{m}\left(j\right),y_{m}\left(k\right)\right]\leq r$$
where 1 ≤ *j* ≤ *N − m* and 0 ≤ *k* ≤ *m* − 1. The number of vectors,$\;B_{m}$, which are significantly different are calculated and normalized as:(2)$$B_{m}=\frac{B}{N-m-1}$$

Repeating the process for vectors of length *m* + 1, denoting the corresponding value as *B_m+1_*, SampEnt is then calculated by:(3)$$SampEnt\left(m,r,N\right)=-ln\lceil \frac{B_{m+1}}{B_{m}}\rceil$$

The size of the time series, *N*, will affect the value of SampEnt with interactions between tolerance *r* and vector length *m* reported [39]. Yentes et al. (2013) observed that a minimum of 200 step parameter data points are needed for SampEnt to stabilize [40]. However, too large values of *N* may affect the calculation of entropy by increasing drift or nonstationarities in the data [41].

SampEnt is dependent on values *m* and *r* (Figure 4), therefore, optimal selection is essential. Values of SampEnt, based on our experimental data, decreases with increasing *r*, in agreement with previous studies [40]. In theory, the accuracy of the entropy estimate relies upon the confidence of the conditional probability and thus, by definition improves with an increased number of matches [41]. Although a short template, *m*, and wide tolerance *r* would optimize the number of matches, the entropy estimate is defined in the limit as *m* tends to infinity and *r* tends to zero [42], therefore, a suitable compromise must be made. A short template length of, for example m = 1, results in a loss of information as patterns involving longer lengths may not be detected. As the tolerance r increases, there are increased matches and sample entropy tends to zero, however, features may be missed. One method of determining *m* is to fit an autoregressive (AR) model to the data, whereby the optimal order of the model provides a lower bound for *m*. If the data came from an autoregressive model of order *p,* AR(*p*), then *m* ≥ *p* [42]. The tolerance, *r*, is here defined as the multiple of the standard deviation of the data [38]. One method for identifying considers the length of the confidence interval of the conditional probability estimate. The conditional probability, *CP*, is an estimate of the probability of a match of length *m + 1* given there is a match of length *m,* i.e., *B_m+1_/B_m_*. If the matches *B_m_* were independent and fixed, the random variable B_m+1_ could be modeled by a binomial distribution and the variance of the conditional probability, *CP(1-CP)/B_m_*. Lake et al. (2002) [42] has shown that the estimate of the variance is: (4)$$\sigma_{CP}^{2}=\frac{CP\left(1-CP\right)}{B_{m}}+\frac{K_{B_{m+1}}-K_{B_{m}}{\left(CP\right)}^{2}}{B_{m}^{2}}$$
where $K_{B_{m}}$ is the number of pairs of matching templates of length *m* that overlap and $K_{B_{m+1}}$ with vectors of length *m* + 1, respectively [38]. The standard error for sample entropy can then be estimated by *σ_CP_ /CP*, equivalent to the relative error of the conditional probability [38]. For large *m* and small *r*, the sample entropy estimate is assumed to be normally distributed and thus, the 95% confidence intervals are defined as: (5)$$-\log\left(CP\right)\pm 1.96\frac{\sigma_{CP}}{CP}$$

The optimal value for *r* is identified by minimizing the quantity: (6)$$max\{\frac{\sigma_{CP}}{CP},\;\;\frac{\sigma_{CP}}{-\log\left(CP\right)CP}\}$$
which is the maximum of the relative error of sample entropy and conditional probability (*CP*) estimate, respectively [42]. 

### 2.7. Determining Nonlinearity

Establishing the presence of nonlinearity a priori is essential for the application of nonlinear tools. The surrogate method offers a statistical approach for identifying nonlinearity with the null hypothesis testing that the original time series is a linear Gaussian stochastic process [43]. The surrogate data was generated using the iterated amplitude-adjusted Fourier transform (IAAFT) algorithm [44], which preserves the amplitude distribution and power spectrum of the original time series. To test at the 1% significance level, 99 surrogates were generated, and the Mann–Whitney rank sum test was applied. 

### 2.8. Statistical Analysis

Nonparametric statistical tests were applied, given the small number of participants in each group (*n* = 5). Mann–Whitney rank sum tests were used to assess the nonlinearity of data. The difference between sample entropy of the HOA and PD group was tested with the Wilcoxon signed rank test. The Kendall rank correlation was used to examine associations between SampEnt and clinical features. The statistical significance was set at *p* < 0.05.

## 3. Results

### 3.1. Demographics 

All ten participants were male, aged 67.8 ± 9.8 and 73.4 ± 8.1 years, for the PD and HOA participants, respectively (Table 1). The UPDRS-III, a measure of motor function, ranged from 19 to 41 (Table 1) indicating mild to moderate motor dysfunction [45]. Table 2 lists the number of strides and the mean stride times for the three time-points.

Figure 5 shows a heatmap of the maximum relative error of either SampEnt or the conditional probability for our data, calculated for a range of values of *m* and *r*. The scale represents the efficiency of the entropy estimate. The maximum value of the efficiency metric is when *m ≤* 4. Selecting the maximum value of *m* = 4 [42], optimum values for *r* lie between 0.325 and 0.375. Values *m* = 4 and *r* = 0.35 were chosen with a conditional probability ≤ 0.05. 

### 3.2. Surrogate Analysis

The surrogate data sets for PD and HOA exhibited significantly higher SampEnt values (*p* < 0.05) than the actual data sets (Table 3), indicating nonlinearity of the stance time real-world data for both groups. 

### 3.3. Sample Entropy

The PD participants had significantly higher SampEnt values (*p* = 0.008) across the three time-points compared to the HOA group (Table 3). There was no significant change in SampEnt between time-points for either the PD or HOA group. There was a greater range in SampEnt for the PD group at TP3 (0.57) compared to TP1 (0.14) (Figure 6, Table 4).

### 3.4. Clinical Features and Sample Entropy

People with PD displayed different changes in progression of the clinical measure of PD motor severity, UPDRS-III, with two people decreasing their score from time-point 1 to time-point 3, one person displaying little change and two people increasing their scores (Figure 7a). The dopaminergic medication level, LEDD, increased for all five participants from time-point 1 to time-point 3 (Figure 7b). There was no significant association between SampEnt and UPDRS-III (Kendall’s *τ* = −0.221, *p* = 0.127). However, a significant correlation was found between SampEnt and LEDD (Kendall’s *τ* = 0.394, *p* = 0.021). 

## 4. Discussion

This is the first application to our knowledge of nonlinear methods to real-world gait data of people with PD over an extended time period. The purpose of this study was to examine the feasibility of applying nonlinear analyses to real-world gait data to identify differences between PD and HOA groups. Additionally, we aimed to investigate if nonlinear gait metrics could detect changes over a 36-month period. We hypothesized that regularity would be lower in the PD group and would decrease over 36 months. Supporting our first hypothesis, people with PD had significantly greater SampEnt, indicating lower regularity than the HOA group (Table 3) although it did not increase over time, contrary to our second hypothesis. However, we did observe that there was a positive association between SampEnt and dopaminergic medication levels, which increased over 36 months for all five people with PD.

The lower gait regularity we observed in the PD group may be explained by the greater number of adjustments needed to overcome the increasing instability resulting from impaired sensorimotor integration [46]. Our results are in agreement with two previous studies, which applied recurrence fluctuation analysis and LRA to determine nonlinear variability [14]. This suggests more random gait dynamics in people with PD. However, Kamath (2015) reported that SampEnt of stride time was lower in people with PD (1.27 ± 0.12) compared to healthy adults (1.71 ± 0.13) [15]. This conflicting result may be explained by their parameter selections (vector length *m* = 3, threshold *r* = 0.15, number of strides *N* = 400), younger control group (39.5 ± 18.5 years), and different walking paradigm [15]. Additionally, Kamath reported a shorter stride time and lower standard deviation for the PD group (1.11 ± 0.05 s) and control group (1.07 ± 0.02 s), reflecting the different types of walking data. 

Although SampEnt was greater for the PD group, we did not observe any increase over 36 months, contrary to our hypothesized decrease in regularity over time there was, however, an increase in range from TP1 to TP3 in the PD group whereas the range of SampEnt for the HOA group decreased. The absence of change in SampEnt may be related to the varying stages of motor dysfunction, indicated by the MDS-UPDRS-III motor score ranging from 19–43 and different patterns of progression of movement disorder, with no consistent increase or decrease in the PD cohort. A study investigating dynamic postural SampEnt similarly observed increased variability in PD compared to HOA but no change over three years [17]. A further consideration is that the dopaminergic medication will modify gait parameters. A positive correlation between SampEnt and the dosage of dopaminergic medication, which increased over thirty-six months, was observed. One interpretation of this association is that LEDD is an indicator of the severity of the disease, therefore, greater SampEnt is associated with greater disease severity. A different interpretation is causal and is that the effect of dopaminergic medication is to increase stride time variability. However, previous studies have reported reduced stride time variability [47] and greater regularity of ankle joint kinematics [48] when in the ON medication state. Interpretation of LEDD is difficult as a recent study reported that people who had a greater motor response to a levodopa challenge test were prescribed higher levels of LEDD and exhibited reduced deterioration in motor function, quantified by the UPDRS-III score, in the eighteen months prior to the levodopa challenge test [49]. A final factor, which may affect gait due to dopaminergic medication is levodopa induced dyskinesia, abnormal movement patterns that develop directly as a result of dopaminergic medication [50]. However, we thresholded the strides for duration, which is likely to minimize inclusion of gait associated with dyskinesias. One major issue, when considering the association between SampEnt and dopaminergic medication is that there was no constraint in medication levels during recording as gait recordings were collected during the entire day when participants would have been in both the ON and OFF states. 

There are several limitations to this study. As this was a feasibility study, data from only a small number of PD and HOA were processed, which reduces the statistical power. The number of strides analyzed, *N*, differed between participants. Although we did not find a correlation between *N* and SampEnt, we did not investigate the interactions between m and r. However, given that the minimum number of strides examined for each participant was over 4600, SampEnt is expected to have stabilized and not to change with the increasing *N*. Future work will include extracting nonlinear parameters from other gait metrics and domains from a larger number of participants. Additional nonlinear measures will be determined such as local dynamic stability or the correlation dimension.

## 5. Conclusions

Analysis of 262,735 strides during real-world walking over a 36-month period indicated significantly lower regularity of stride time in people with Parkinson’s compared to healthy age-matched older adults. The novelty of this study is that we analyzed real-world gait, recorded over seven days, during a 36-month period. The importance of real-world gait data is that not only are changes in internal gait timing systems monitored but also responses to external perturbations assessed as individuals undertake complex walking tasks such as crossing roads, turning corners, talking, and adjusting speed. Additionally, real-world data recorded over several days incorporate circadian variations, different levels of dopaminergic medication reflecting both the ON and OFF states, day to day fluctuations in mood, and variations in daily activities, which may modify gait. The lower regularity of stride time in people with PD may reflect not only deterioration of dopaminergic neurons within the basal ganglia but also changes in other subcomponents such as the postulated spinal central pattern generator circuits controlling gait and the response to external environmental perturbations. Although some variability of gait parameters is desirable, indicating flexibility and ability to adjust to perturbations, too high a variability implies an unstable control system and instability leading to falls. We did not observe an increase in sample entropy with disease duration, which may be due to the small sample size and heterogeneity of the individual participants in addition to the effect of medication. However, we did observe a positive correlation between SampEnt and LEDD, the dopaminergic medication daily dose. Further investigation is needed to determine underlying causes for this association. In summary, applying sample entropy algorithms to real-world data shows potential as a method to differentiate people with PD from HOA. However, it is essential that appropriate vector lengths and thresholds are established, and signals are tested for nonlinearity. Further analyses are essential, involving greater numbers of participants, applying different methods of nonlinear analysis, investigating different gait parameters, and relating the nonlinear measures to functional measures.

## Figures and Tables

**Figure 1 sensors-20-02631-f001:**
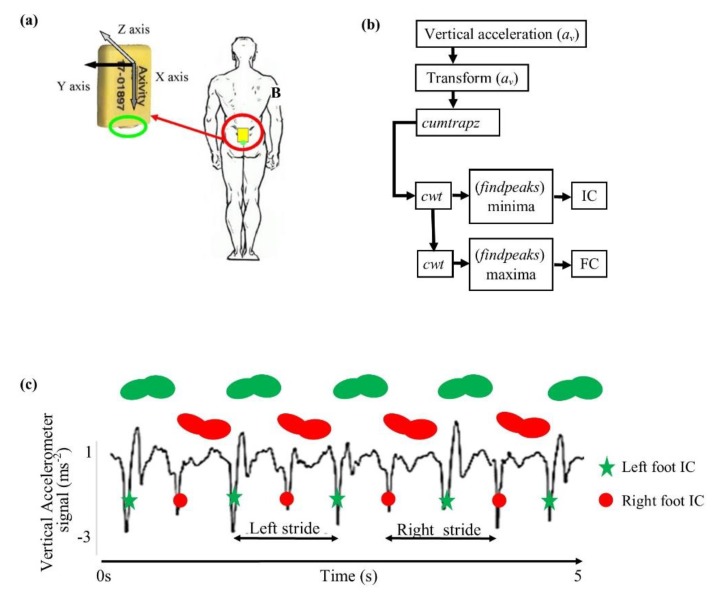
(**a**) Experimental setup: The site of attachment and orientation of the tri-axial accelerometer device on the lower back (L5). In dark grey x (vertical) axis, in black y (mediolateral) axis, and in light grey z (anteroposterior) axis. (**b**) Flowchart of analysis to determine gait events using MATLAB functions *cumtrapz*, *cwt,* and *findpeaks*. IC: Initial contact; FC: Final contact. (**c**) Vertical component of accelerometer signal with gait events and stride cycles indicated.

**Figure 2 sensors-20-02631-f002:**
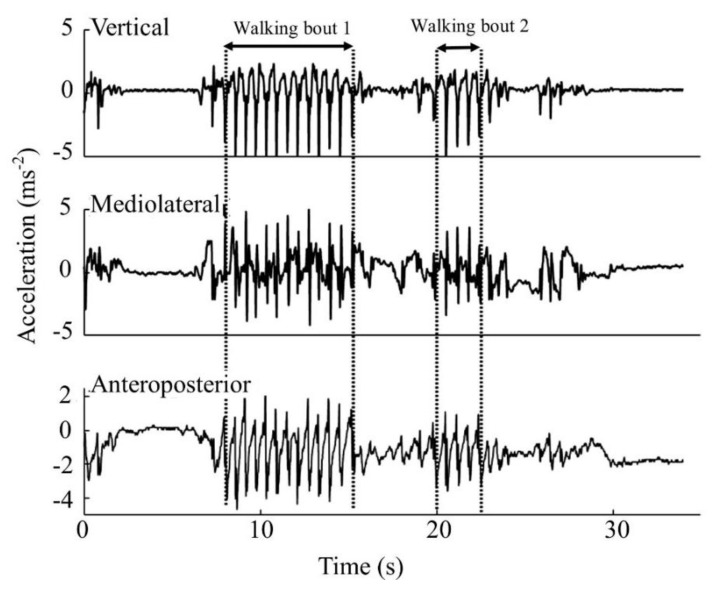
Output data of a body worn monitor for one participant showing vertical, mediolateral, and anteroposterior acceleration. A longer walking bout 1 and shorter walking bout 2 are indicated.

**Figure 3 sensors-20-02631-f003:**
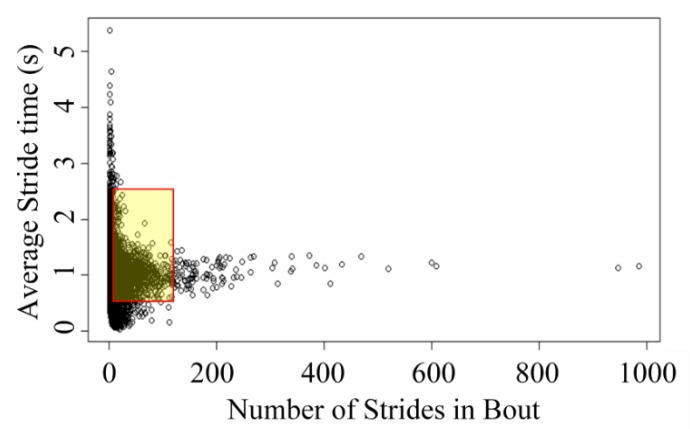
Mean stride time for walking bouts of length 1–1000 for a Parkinson’s disease participant, indicating dependence of stride time on the duration of walking bout. Shaded area indicates included bouts.

**Figure 4 sensors-20-02631-f004:**
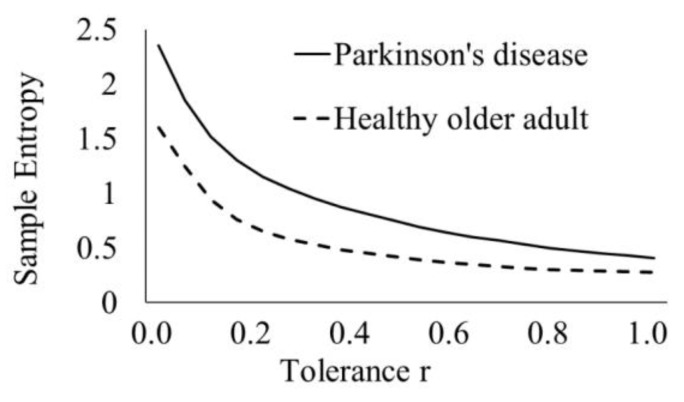
Sample entropy of stride time as a function of tolerance *r* for averaged *m* = 1*,*2*,*3*,*4 for a participant with Parkinson’s disease (PD1) (solid line) and a healthy older adult (HOA1) (dashed line).

**Figure 5 sensors-20-02631-f005:**
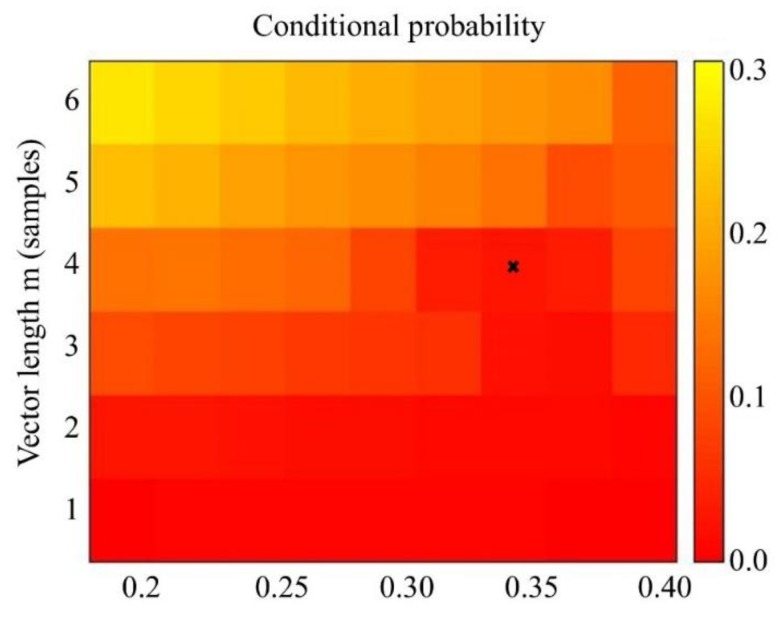
Maximum relative error of the sample entropy or conditional probability, as a function of *m* and *r*. Optimal parameters *m* = 4, *r* = 0.35.

**Figure 6 sensors-20-02631-f006:**
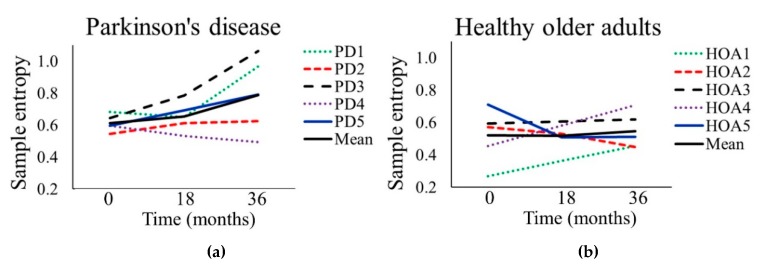
Sample entropy of stride time over a 36-month period for (**a**) five people with Parkinson’s disease (PD1–PD5) and (**b**) five healthy older adults (HOA1–HOA5).

**Figure 7 sensors-20-02631-f007:**
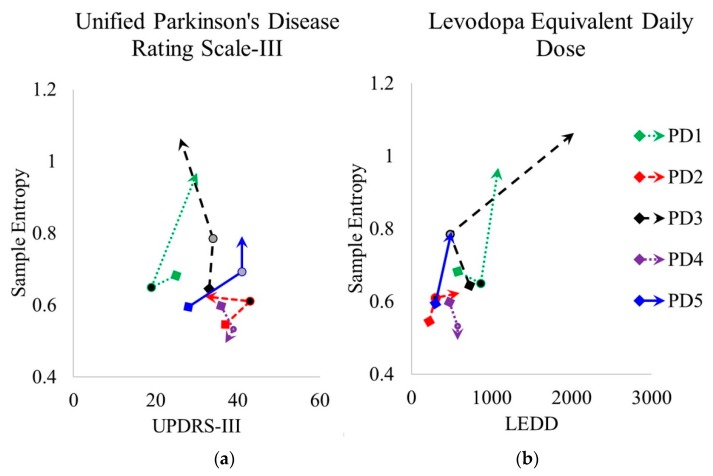
Sample entropy of stride time for time-point 1 (0 months), time-point 2 (18 months), and time-point 3 (36 months) for five people with Parkinson’s (PD1–PD5) plotted against (**a**) Unified Parkinson’s Disease Rating Motor Scale (UPDRS-III) and (**b**) levodopa equivalent daily dose (LEDD).—values at 0 months;—values at 18 months;—values at 36 months.

**Table 1 sensors-20-02631-t001:** Participant characteristics and clinical motor scores for a group with Parkinson’s disease.

ID	Age	Height	Mass	BMI	MDS-UPDRS-III			LEDD		
	(yrs)	(m)	(kg)	(kg/m2)	TP1	TP2	TP3	TP1	TP2	TP3
**PD1**	57.6	1.78	81.8	25.8	25	19	30	580	865	1081
**PD2**	61.3	1.73	72.8	24.3	37	43	32	220	300	600
**PD3**	65.2	1.76	112.4	36.3	33	34	26	730	483	2031
**PD4**	80.1	1.72	68.6	23.2	36	39	37	475	575	575
**PD5**	76.5	1.74	89.8	29.7	28	41	41	300	400	500
***Mean (SD)***	***68.1 (9.7)***	***1.74 (0.02)***	***85.1 (17.3)***	***27.9 (5.3)***	***32(5)***	***35 (10)***	***33 (6)***	***461 (207)***	***525 (216)***	***957.4 (643)***
**HOA1**	77.0	1.75	76.4	24.9	-	-	-	-	-	-
**HOA2**	61.6	1.75	83.4	27.2	-	-	-	-	-	-
**HOA3**	73.8	1.84	110.6	32.7	-	-	-	-	-	-
**HOA4**	69.6	1.76	80.6	27.9	-	-	-	-	-	-
**HOA5**	84.0	1.74	82.0	27.1	-	-	-	-	-	-
***Mean (SD)***	***73.2(8.3)***	***1.77(0.04)***	***86.6(13.7)***	***28.0(2.9)***	**-**	**-**	**-**	**-**	**-**	**-**

BMI: Body mass index; MDS-UPDRS-III: Movement Disorders Society-Unified Parkinson’s Disease Rating Scale Part III; TP1: Time-point 1 = 0 months; TP2: Time-point 2 = 18 months; TP3: Time-point 3 = 18 months; LEDD: Levodopa equivalent daily dose; PD: People with Parkinson’s disease; HOA: Healthy older adults.

**Table 2 sensors-20-02631-t002:** Average number of strides analyzed, average strides per bout, and average stride time at time-points 0, 18, and 36 months.

Group	Time (Months)	Total Strides	Strides Per Bout	Stride Time (S)
PD	0	7244	29.7 ± 7.8	1.31 ± 0.21
PD	18	8502	30.7 ± 6.9	1.27 ± 0.17
PD	36	9473	31.7 ± 7.8	1.26 ± 0.17
HOA	0	7563	23.3 ± 5.2	0.93 ± 0.11
HOA	18	10386	29.9 ± 7.2	1.28 ± 0.16
HOA	36	9379	29.0 ± 7.1	1.31 ± 0.18

PD: People with Parkinson’s disease; HOA: Healthy older adults.

**Table 3 sensors-20-02631-t003:** Sample entropy of original and surrogate time series for people with Parkinson’s disease and healthy older adults.

Group	Time Series	*p*-Value *	Surrogate Time Series	*p*-Value **
PD	0.65 ± 0.09	0.008	1.31 ± 0.06	5.95 × 10^−5^
HOA	0.55 ± 0.11		1.27 ± 0.09	5.95 × 10^−5^

PD: Person with Parkinson’s disease; HOA: Healthy older adults; * One-tailed Wilcoxon signed-rank test p-value for PD and HOA SampEnt; ** Mann–Whitney rank sum test p-value.

**Table 4 sensors-20-02631-t004:** Sample entropy for five participants with Parkinson’s disease and five older adults at TP1, TP2, and TP3.

ID	TP1		TP2		TP3	
	N	SampEnt	N	SampEnt	N	SampEnt
PD1	5618	0.68	11,647	0.65	10,721	0.97
PD2	6844	0.55	4691	0.61	7878	0.62
PD3	6800	0.65	6860	0.79	6761	1.06
PD4	7988	0.60	9668	0.53	12,999	0.50
PD5	8970	0.60	9646	0.69	9007	0.79
***Mean*** ***± SD***	***7244*** ***± 1278***	***0.61*** ***± 0.05***	***8502*** ***± 2729***	***0.65*** ***± 0.09***	***9473.2*** ***± 2455***	***0.79*** ***± 0.24***
HOA1	7574	0.27	13366	0.36	12094	0.46
HOA2	6545	0.57	8088	0.53	7204	0.45
HOA3	8716	0.59	9527	0.61	10081	0.62
HOA4	8179	0.46	9156	0.58	7208	0.71
HOA5	6805	0.71	11792	0.51	10308	0.51
***Mean*** ***± SD***	**7564 ± 911**	**0.52 ± 0.17**	**10386** **± 2144**	**0.52 ± 0.10**	**9379 ± 2131**	**0.55 ± 0.11**

PD: People with Parkinson’s disease; HOA: Healthy older adults; TP1: Time-point 1; TP2: Time-point 2 = TP1 + 18 months; TP3: Time-point 3 = TP2 + 18 months. N: Number of stride cycles analyzed.

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
