# Peer review of "Entropy of Real-World Gait in Parkinson’s Disease Determined from Wearable Sensors as a Digital Marker of Altered Ambulatory Behavior"

_sensors, 2020, doi:10.3390/s20092631_

Round 1

Reviewer 1 Report

It is very important to analyze the gait characteristics of Parkinson patients. The author analyzed about ten persons’.data. The paper needs to be revised in the following aspects. 1. Please add figure to illustrate how to do the experiment in details, and the position of the sensors should be described with figure. You can refer to the paper with the name "The Effect of Treadmill Walking on Gait and Upper Trunk through Linear and Nonlinear Analysis Methods" 2. More equations should be used to illustrate the proposed method clearly. 3. Current figures without unit, which should be revised. 4. Statistical methods should be used to analyze the results.

Reviewer 2 Report

This paper studied whether sample entropy derived from gait stride time can be used to distinguish PD patients with the healthy elderly. Real-world gait data were collected by accelerometer placed on the low back for 7 days and gait metrics were extracted. Results showed that PD patients had higher stride time SampEnt than healthy older adults, indicating that PD patients have greater gait dynamic variability. The research is well designed and executed and the manuscript is well written. Before publication, following modifications needed to be addressed.

Line 211 Table 3 è should be Table 4 ?

Is the p-value shown at the rightmost column of Table3, i.e., 0.002. calculated based on the surrogate time series collected at a particular time point or all three TPs?

As the sample size N=5 is small, it is recommended that the authors address the statistical power of such sample size.

In Table 2, is there any statistical significance between the PD group and HOA group at each TPi, i=1, 2, 3, supposed that the sample size is addressed. If so, can we use No. of strides (in addition to Sample Entropy) as the bio-maker to distinguish PD with HOA?

Reviewer 3 Report

General comments:

The study investigates the feasibility of body-worn accelerometer signals in combination with nonlinear analysis methods to detect differences in walking behavior in patients with Parkinson’s disease and healthy participants. Furthermore, the study aims to evaluate the capability of the setup to detect longitudinal changes in behavior, as it includes gait recordings from three time-points across 3 years. The strength of the study is the innovative approach to extract functional meaningful outcomes from unsupervised real-world gait recordings and the methodology to define sample entropy input parameters. The weakness of the study is the very small sample size, and the resulting inability to draw clear statistical conclusions in terms of validity of the approach with respect to the clinical assessment and longitudinal changes of walking behavior in patients. In part, this leads to statements that inconsistent and unsupported by the data.

My major concern is the inconsistent statistical approach. The sample size is very small with five subjects in each group. The authors find significant differences when comparing healthy subjects to patients, however, the data does not allow to display changes of walking behavior in patients over time, nor a comparison of walking performance and the longitudinal clinical UPDRS scores. Possibly, because the study is underpowered. I don’t think the low sample size is a problem per se, but the authors should decide if they run traditional statistical methods and interpret results accordingly. Alternatively, they can choose to simply describe the findings, knowingly that it is difficult to make general inferential statements with such low samples.

However, the overall approach and the presentation of the study are very good. The approach is innovative and very much needed to allow for a better real-world observation of gait function in clinical populations. Therefore, I recommend accepting the study after minor revisions.

Specific comments:

L4: I am troubled with the title, as it indicates “disease progression”. Since no progression was observed (or at least I could not tell from how the results are presented), it is a bit difficult to speak to that. I recommend to either formulate the title as a question or to remove “disease progression” and probably use more generic language such as: “…digital marker of insufficient ambulatory behavior.”

L19 & 23: I am not sure what “dynamic variability” is. Does it mean “gait variability” (as compared to static, postural sway type variability) or variability that is considerate of the temporal dynamics of the system (as compared to linear variability measures)? Anyway, since SampEnt is already introduced, I recommend being more specific here. SampEnt quantifies “regularity” and I recommend using this term here instead.

L88: “real-world” appears twice here.

L89: Does not the “H” in HOA stand for “healthy”?

L95ff.: Can this be stated a bit more concise?

L104: I find this confusing. How were the cognitive testing results included in the analysis? I thought the measurements were taken outside the laboratory? How are these gait and balance assessments included in the analysis, as compared to the real-world measurements?

L104: I assume that during this session also the UPDRS was measured (which should be mentioned). During these measurements, were patients ON medication or OFF? In general, information about the treatments that patients received (pharmacological or others) should be added. I assume that for the real-world recordings it was not controlled for if patients were walking ON or OFF, correct? This would be good to include in the discussion.

L134: Since longer consecutive bouts are usually preferred during the nonlinear analysis of gait, please provide a reason why bouts of longer than a minute were not considered.

L135: Please explain, what happened when a stride outside the inclusion range was found within a walking bout (i.e. consecutive stride time vector)? Was this value removed, or replaced (with what value?) or was the entire bout discarded?

L149: This figure is presented in the context of “theoretical” consideration regarding m, r and sample entropy. Please indicate in the figure caption if the data presented here is based on the data or if this is a theoretical model? If this is based on the data, please describe how it compares to the results of Yentes et al. (Annals of Biomedical Engineering 2013; 41: 349-365).

L152: I recommend providing the reference to this statement.

L155: I recommend adding an explanation of what the unwanted consequences of too large r and too short m selections are. It makes it easier for the reader to follow the need to identify appropriate values for m and r.

L166: I assume the “m” should be in italics.

L188: One aspect that has not mentioned before is that SampEnt is also depending on N (Yentes et al. GAIPOS 2018; 60: 128-134). Please add information regarding the average length of N in the analysis? Furthermore, there were a different number of strides in each TP, I assume N were different too. Has it been analyzed, how this might have affected the SampEnt results? I recommend (at the least) adding the information about the size of N here and then to add this aspect to the limitations section. Better would be to investigate the effect in an additional analysis or alternatively to use an equal N-long time-series per each subject.

L192: It would be nice if the color code of the heatmap could be scaled in a way to display the maximum efficiency point. Right now, it looks as if all r at m = 4 have the same probability.

L210: This sentence highlights my main concern. Either the results are non-significant, then SampEnt did not increase over the measurement period. Or, because of the low sample size that permits drawing clear statistical conclusions, the data is presented only in a descriptive way. To me, the whole point of inferential statistics is to make inferences about the estimated population behavior.

L223: Here is the problem again, in L210 it was mentioned that SampEnt increased over the 36-month period, now it is stated that it did not.

L268: A formal analysis would be necessary to make a conclusion regarding the association between SampEnt and clinical scores (i.e. disease progression). Also, I don’t think that the data supports the observation of disease progression. How was this measured, in the UPDRS? See my previous comments.

Round 2

Reviewer 1 Report

  The author has revised the manuscript in some parts. However, there are also some aspects that need to be improved. Fig. 1 and Fig. 2 are not clear. It is better to change the color in Fig. 5.

  To analyze the experimental data in details, it is better for the author to draw figure like Fig. 3 (a) in the paper with the title "The Effect of Treadmill Walking on Gait and Upper Trunk through Linear and Nonlinear Analysis Methods", and refer to it. Some of the references in the paper are more than twenty years old (1991, 2000 etc.), it is better to change to the latest one.

  Even the author has add the name in some figures, such as Fig.6, there is still no unit. The readers have to guess it. For the Statistical Analysis, just giving the  statistical significance byt p is not enough,  it is better to draw Box-plot figure to illustrate it clearly. The Fig. 7 cannot give the difference clearly. Please change the description format.

Round 3

Reviewer 1 Report

The author has revised the manuscript according to the requirment. It can be accepted.